# Alterations of the Mucosal Immune Response and Microbial Community of the Skin upon Viral Infection in Rainbow Trout (*Oncorhynchus mykiss*)

**DOI:** 10.3390/ijms232214037

**Published:** 2022-11-14

**Authors:** Mengting Zhan, Zhenyu Huang, Gaofeng Cheng, Yongyao Yu, Jianguo Su, Zhen Xu

**Affiliations:** 1Department of Aquatic Animal Medicine, College of Fisheries, Huazhong Agricultural University, Wuhan 430070, China; 2State Key Laboratory of Freshwater Ecology and Biotechnology, Institute of Hydrobiology, Chinese Academy of Sciences, Wuhan 430072, China; 3Laboratory for Marine Biology and Biotechnology, Qingdao National Laboratory for Marine Science and Technology, Qingdao 266071, China

**Keywords:** rainbow trout, infectious hematopoietic necrosis virus (IHNV), mucosal immunity, skin microbiota, RNA-Seq, 16S rRNA sequencing

## Abstract

The skin is the largest organ on the surface of vertebrates, which not only acts as the first line of defense against pathogens but also harbors diverse symbiotic microorganisms. The complex interaction between skin immunity, pathogens, and commensal bacteria has been extensively studied in mammals. However, little is known regarding the effects of viral infection on the skin immune response and microbial composition in teleost fish. In this study, we exposed rainbow trout (*Oncorhynchus mykiss*) to infectious hematopoietic necrosis virus (IHNV) by immersion infection. Through pathogen load detection and pathological evaluation, we confirmed that IHNV successfully invaded the rainbow trout, causing severe damage to the epidermis of the skin. qPCR analyses revealed that IHNV invasion significantly upregulated antiviral genes and elicited strong innate immune responses. Transcriptome analyses indicated that IHNV challenge induced strong antiviral responses mediated by pattern recognition receptor (PRR) signaling pathways in the early stage of the infection (4 days post-infection (dpi)), and an extremely strong antibacterial immune response occurred at 14 dpi. Our 16S rRNA sequencing results indicated that the skin microbial community of IHNV-infected fish was significantly richer and more diverse. Particularly, the infected fish exhibited a decrease in Proteobacteria accompanied by an increase in Actinobacteria. Furthermore, IHNV invasion favored the colonization of opportunistic pathogens such as *Rhodococcus* and *Vibrio* on the skin, especially in the later stage of infection, leading to dysbiosis. Our findings suggest that IHNV invasion is associated with skin microbiota dysbiosis and could thus lead to secondary bacterial infection.

## 1. Introduction

The skin of vertebrates provides an effective physical and immune barrier that guards the host against external stimuli and pathogen invasion [1]. Unlike mammals, the skin of teleost fish is considered a mucosal tissue, which contains living epithelial and mucus-secreting cells that are in direct contact with the water environment [2]. Furthermore, teleost skin has been found to possess skin-associated lymphoid tissue (SALT) and perform immune functions similar to those of the gut [3]. Recent studies have demonstrated that fish skin is an important inductive and effector site to resist parasitic and bacterial invasion, which is able to elicit strong innate and adaptive immune responses [4,5]. Moreover, due to their direct contact with their aqueous environment, the skin of teleost fishes is naturally colonized by millions of highly abundant and diverse microorganisms [6]. Some symbiotic microbiota that colonize the mucosal surface are beneficial and participate in key physiological processes such as host development [7], pathogen resistance [8,9,10,11], and nutrition [12]. However, the skin also harbors various potentially opportunistic pathogens whose abundances might increase when the host is invaded by other pathogens or is in a state of microbial imbalance, thus resulting in serious diseases [13]. To date, very few studies have characterized the effects of pathogen invasion on the immune responses and microbial community in teleost fish skin.

Pathogen invasion often disrupts the microbial balance of the host, and this state of dysbiosis can trigger strong immune responses or serious disease [14,15]. The relationship between host immunity and pathogen-induced microbial community changes in the skin has been extensively studied in mammals [16,17]. For instance, patients infected with *Leishmania braziliensis* develop skin dysbiosis, which is characterized by increases in the abundance of *Staphylococcus* and/or *Streptococcus*, both of which can induce inflammation [18]. Some studies in fish have also evaluated the association between pathogens, skin immunity, and microbial community structure [19]. Upon infection with *Ichthyophthirius multifiliis*, the abundance of the predominant phylum Proteobacteria in rainbow trout skin significantly decreased, whereas the abundance of the pathogenic Flavobacteriaceae increased, which coincided with a marked upregulation of antimicrobial genes [4]. In recent years, there has been an increasing interest in the effects of viral infection on the microbial composition of fish skin. For example, salmonid alphavirus infection in Atlantic salmon can lead to skin dysbiosis, which may render the host more susceptible to secondary bacterial infection [20]. However, very few studies have simultaneously characterized the effects of viral infection on skin immune responses and microbial changes.

Rainbow trout (*Oncorhynchus mykiss*) is not only an economically relevant cold-water fish species but is also a widely studied fish model [21,22]. The incidence of infectious disease outbreaks in rainbow trout farms has steadily increased in recent years due to the intensification of trout aquaculture [23], among which infectious hematopoietic necrosis disease (IHN) has caused considerable economic losses [24]. IHN is one of the bottlenecks restricting the development of the world’s cold-water fish industry, whose causative agent is infectious hematopoietic necrosis virus (IHNV) [25,26]. This virus belongs to the genus Novirhabdovirus of the Rhabdoviridae family and poses a major threat to juvenile salmon and trout [27,28]. Fish infected with this virus show several characteristic clinical symptoms including abdominal distension, exophthalmia, darkening of the skin, abnormal behavior, anemia, and pale gills [29]. Although previous studies have shown that IHNV invasion induced strong immune responses of the digestive tract and swim bladder of rainbow trout [30,31], very little is still known regarding the effects of IHNV infection in juvenile trout skin.

To understand the immune responses and microbial changes in fish skin upon viral infection, IHNV infections were induced by immersing the trout in an IHNV-containing bath. Our results showed that IHNV infection caused severe tissue damage in rainbow trout skin and led to strong immune responses. The results of transcriptome sequencing further demonstrated that antiviral and antimicrobial immunity were activated at the early and late stages of infection, respectively. Moreover, IHNV challenge induced notable changes in the microbial communities of the skin, causing increased abundances of various opportunistic pathogenic bacteria, which might lead to secondary bacterial infection. These results demonstrated that IHNV infection led to microbial dysbiosis and induced an antibacterial immune response in the skin. Our results provide some insights into the effects of viral infection on the immune response and microbial community of teleost skin, thus serving as a theoretical basis for the development of effective strategies to control IHN disease in aquaculture.

## 2. Results

### 2.1. IHNV Successfully Invaded the Skin of Rainbow Trout

After exposing rainbow trout to IHNV by immersion infection, we collected skin and head kidney samples at 1, 4, 7, 14, 21, and 28 dpi (Figure 1a). About 4 to 7 days after IHNV infection, obvious symptoms could be observed on the diseased fish (Figure 1b), including slow swimming, darkening of the skin, exophthalmia, and pale gills. Through daily observation, we found the overall mortality rate of the fish population is about 30%, and most deaths occurred from 5 to 10 dpi (Figure 1c). qPCR results showed that the viral load in the skin significantly increased at 4, 7, and 14 dpi, then recovered at 21 and 28 dpi (Figure 1d). Similar viral load changes were also observed in the head kidney, one of the target tissues of IHNV (Appendix A). Additionally, using anti-IHNV-N mAb, we detected obvious fluorescence signal of IHNV in the epidermis, dermis, and even muscle layer of fish skin (Figure 1e) (isotype control antibody staining for IHNV detection is shown in Appendix A). Moreover, we collected the supernatant of skin tissue homogenates from the 7 dpi trout and added them to the well-cultured EPC cells, and obvious plaques could be observed after 48 h of culture (Figure 1f). Taken together, these results demonstrated that IHNV had successfully invaded the fish skin.

### 2.2. IHNV Infection Caused Significant Histopathological Changes and Strong Innate Immune Responses in Rainbow Trout Skin

Paraffin sections of skin samples at different time points were stained with hematoxylin–eosin (H&E) and Alcian blue (AB) to detect the pathological changes in skin caused by IHNV. H&E staining indicated that the structure of the skin epidermis was seriously damaged, and epithelial cells fall off at 7 dpi (Figure 2a). Statistical analysis showed that the epidermis of the infected fish was obviously thinner, especially at 7 and 14 dpi, and then gradually recovered (Figure 2b). Similarly, AB staining showed that the number of mucus cells in the infected fish skin decreased significantly at 4, 7 and 14 dpi and then recovered (Appendix A). The above results showed that IHNV invasion could cause severe skin lesions in rainbow trout. To further investigate changes of immune-related gene expression at different time points after infection, 9 immune-related genes were examined by using qPCR (Figure 2c–k), including antiviral genes (mx1, ddx58, ifih1, stat1, dhx58, and tlr3), complement component (c3), interleukin (il6), and antibacterial genes (cath1) (qPCR primer sequences are shown in Appendix A). The results showed that IHNV infection resulted in strong antiviral immune responses in trout skin, especially in the early stage of infection (4 and 7 dpi). The expression of c3 and il6 also increased significantly. The fish of 14 dpi group witnessed a higher expression of cath1, indicating that antibacterial immune responses may occur at 14 dpi.

### 2.3. Transcriptome Profiling in the Skin of Rainbow Trout after Infection with IHNV

To further explore the changes of immune-related genes and pathways from the transcriptome level, we performed RNA-Seq of skin samples from 4 and 14 dpi trout. An overview of the reads and quality filtering of the RNA-seq libraries is presented in Appendix A. After filtering the sequences with thresholds and removing the repeats, we obtained 43,151,119; 45,189,997; and 44,261,967 reads in control 4 dpi; and 14 dpi groups, respectively. Prior to DEG analysis, principal component analysis (PCA) was used to assess the biological variability and to visualize the distribution of various groups (Figure 3a). As shown in the plot, the transcriptional profile of the 4 dpi group clustered separately from that of the control and 14 dpi group, while the 14 dpi group overlapped with the control group, indicating that the 4 dpi group showed changes in transcriptome level that were more pronounced compared to that of the 14 dpi group. For the differential gene expression analysis, *p* (adj) < 0.05 and |log2FoldChange| > 1 were set as a threshold level to retrieve the DEGs. Information on differentially expressed genes (DEGs) is provided in Appendix A, which is a Venn diagram showing the number of genes that are differentially expressed at 4 and 14 dpi compared to control group (Figure 3b). A total of 2406 (4 dpi) and 478 (14 dpi) genes were significantly dysregulated after IHNV infection, of which 1337 and 142 genes were upregulated, whereas 1069 and 336 genes were downregulated at 4 and 14 dpi, respectively. Then, we analyzed and compared the DEGs at 4 and 14 dpi and selected 20 representative immune-related genes (Table 1). We found that a strong antiviral immune response occurred in the skin at 4 dpi after IHNV infection (vig1, mx2, trim39, ifnar2, irf7, irf8, gig2e, ddx58, stat1b, dhx58), and the expressions of inflammatory factors (il4i1) and chemokines (ccl19a.1) were also significantly upregulated. At 14 dpi, the expression of antibacterial-related genes (hamp, saa5, cd209, cath, lyz2) was stronger, and the expression of complement-related genes (c1q1, c4b) was also significantly up-regulated. The results indicated that the skin of rainbow trout initiates a strong antiviral and innate immune response to fight against IHNV invasion in the early stage of infection (4 dpi). Perhaps more importantly, significantly high expression of antimicrobial genes in the 14 dpi group revealed that IHNV infection may cause secondary bacterial infections.

As a mean of validation of RNA-seq, the RNA samples used for the sequencing database construction were used to detect the expression of 8 DEGs by qPCR (Appendix A). As shown in the bar chart, the qPCR results matched well with the RNA-seq data, which means that the RNA-seq results were reliable. The qPCR primers used for validation are shown in Appendix A.

### 2.4. KEGG Pathway and GO Enrichment Analysis of DEGs

To identify the signaling pathways and biological processes regulated by IHNV infection, we performed KEGG pathway analysis and GO enrichment analysis. KEGG analysis of DEGs in the 4 dpi group identified many immune-related pathways including “Cytokine-cytokine receptor interaction” (59 DEGs), “Nod-like receptor signaling pathway” (44 DEGs), “C-type lectin receptor signaling pathway” (31 DEGs) and “Toll-like receptor (TLR) signaling pathway” (29 DEGs), which play a critical role in recognition of pathogens and activation of innate immune response (Table 2). In addition to immune-related pathways, those involved in cell signal transduction and interaction, energy supply and protein processing were also enriched in the 4 dpi group. In the 14 dpi group, only three KEGG pathways were significantly enriched: “ECM-receptor interaction” (18 DEGs), “Focal adhesion” (23 DEGs) and “Proteasome” (7 DEGs) (Table 3). Furthermore, we focused on the biological processes in GO enrichment analysis, which were related to signal transduction, host defense response and immune response. The bubble chart (Figure 3c,d) indicates that various innate immune-related pathways were significantly enriched in both the 4 and 14 dpi groups, of which type I interferon response was mainly enriched at 4 dpi, while response to bacterium was mainly enriched at 14 dpi.

### 2.5. IHNV Infection Caused Changes in the Composition of Trout Skin Microbial Community

In order to explore the impacts of IHNV infection on the skin microbiota of rainbow trout, we performed 16S rRNA sequencing (after preliminary analysis, two samples in the control group and two samples in the 4 dpi group were excluded from all subsequent analyses due to their particularly large individual differences). After sequencing on Illumina NovaSeq 6000 platform in pair-end, 1,654,536 raw sequences were obtained. After quality filtering, denoising and chimera removal, a total of 1,134,986 assembled high-quality sequences were retained for further analysis (360,677 in control group, 358,508 in 4 dpi group and 415,801 in 14 dpi group). Information on relative abundance of taxa at all levels (Top 50) is provided in Appendix A. Next, the alpha diversity level of each sample was evaluated according to the distribution of ASVs in different samples. Chao1 index and Shannon diversity index were employed to estimate the alpha diversity (Figure 4a). The box plot showed that compared with the control group, the richness and diversity of skin microbiota in the 4 and 14 dpi groups both were significantly increased. To visualize the community similarity, the non-metric multi-dimensional scaling (NMDS) ordination was performed on Bray–Curtis dissimilarity (Figure 4b). The NMDS showed distinct clusters for three groups of samples: the bacterial community profiles of 14 dpi group scattered to the right of the NMDS plot, control group scattered to the left of the NMDS, and 4 dpi group localized between the above two groups. Significant *p* values of permutation multivariate analysis of variance (Permanova) (Pseudo-F = 5.29; *p* = 0.001) and analysis of similarities (Anosim) (R = 0.58; *p*  <  0.001) between groups emphasize the differences in microbial community structure (Appendix A). These data suggest that IHNV infection causes microbial dysbiosis in the 4 and 14 dpi groups, and that of the 14 dpi group was more significant.

To identify bacterial community structure and the change in bacterial abundance, we examined the average relative abundance of bacterial phyla in the skin samples of the three groups (Figure 4c). The stacked bar chart showed the top 10 most abundant bacteria phyla. Proteobacteria, Actinobacteria, Firmicutes and Bacteroidetes were the four most abundant phyla in all skin samples, which accounted for 94.9% in the control group, 83.1% in the 4 dpi group, and 77.5% in the 14 dpi group. Proteobacteria, the most predominant phylum in all groups, accounted for 78.6 ± 4.2% in the control group and significantly dropped to 51.5 ± 4.1% (*p* = 0.0043) in the 4 dpi group and 43.2 ± 3.4% (*p* = 0.0007) in 14 dpi group. The abundance of Actinobacteria significantly increased from 13.4 ± 2.7% in control fish to 25.8 ± 3.4% (*p* = 0.0152) and 28.2 ± 3.4% (*p* = 0.0027) in the 4 and 14 dpi groups, respectively. At the order level, the abundance of potentially pathogenic taxa increased after IHNV infection, including members of the orders Actinomycetales and Vibrionales (Figure 4d). The abundance of Vibrionales increased from 13.9 ± 3.4% in the control group to 18.1 ± 1.1% (*p* = 0.3874, not significant) in the 4 dpi group and 25.4 ± 2.4% (*p* = 0.02) in the 14 dpi group. At the genus level, we performed a clustering heatmap to visualize the bacterial communities (Figure 4e). The top 20 abundant genera further showed that the 14 dpi group had a greater abundance of potentially pathogenic taxa such as Rhodococcus, Vibrio, Pseudoalteromonas, and Uruburuella. As for the 4 dpi group, potentially pathogenic bacteria Rhodococcus also increased significantly. These results all indicated that after IHNV infection, the taxa and abundance of potentially pathogenic bacteria in the microbiota on the skin mucosa increased significantly, and the degree of microbial dysbiosis was higher at 14 dpi than that of 4 dpi. (Significant differences of different taxa are shown in Appendix A).

### 2.6. The Differentially Abundant Taxa Enriched in Skin Microbiota in 4 dpi Group and 14 dpi Group after IHNV Infection

To identify the specific bacterial taxa and biomarker associated with IHNV infection, LEfSe (LDA Effect Size) analysis of the skin microbiota between the control and infected communities was performed. The structure and predominant bacteria of microbiota in the control group, 4 dpi group and 14 dpi group were represented in the cladogram (Figure 5a). The difference in microbiota from phylum to genus level was identified via LDA score. Compared with the control group, most of the specific taxa fell in Thermis and Actinobacteria in the 4 and 14 dpi groups, respectively. At the genus level, IHNV invasion in trout skin significantly increased the levels of some opportunistic pathogens, including Chryseobacterium and Flavobacterium in the 4 dpi group and Vibrio, Rhodococcus, and Uruburuella in the 14 dpi group. Additionally, we performed scatter diagrams to illustrate the changes in the differentially abundant taxa. An obligate aerobic and Gram-positive genus, Rhodococcus, increased significantly in the 4 dpi (*p* = 0.0152) and 14 dpi groups (*p* = 0.0127) (Figure 5b). The genus Vibrio was more abundant in the 14 dpi group (*p* = 0.02) (Figure 5c). In accordance with this, the species Vibrio metschnikovii (*p* = 0.0293) increased significantly (Figure 5f). In addition, the relative abundance of the opportunistic pathogens Pseudomonas (*p* = 0.0293) (Figure 5d) and Pseudoalteromonas (*p* = 0.02) (Figure 5g) increased significantly at 14 dpi. In addition to the above significantly increased taxa, the abundance of Acinetobacter decreased significantly in the 14 dpi group (Figure 5e). Combining the results above, the invasion of IHNV disrupts the balance of the microbial ecosystem on the trout skin mucosa and resulted in a significant increase in opportunistic pathogens.

## 3. Discussion

The skin of teleosts is a living organ that protects fish against mechanical trauma, as well as external stimuli and pathogens. Due to its direct contact with the water environment, this organ harbors millions of diverse microorganisms that play a crucial role in modulating the immune responses of the skin. Pathogen invasion is known to disrupt the microbial balance of the skin. However, little is known regarding the effect of pathogen-induced dysbiosis on the immune responses of fish skin. Here, we employed an IHNV infection model in rainbow trout to reveal the interactions between immune response, skin microbiota, and the invading virus.

To mimic natural infections, we infected the fish by immersing them in an IHNV-containing bath. The most severe clinical symptoms of the diseased fish were observed from approximately 4 dpi to 7 dpi, including slow swimming, loss of appetite, darkening of the skin, pale gills, and exophthalmia, all of which were generally consistent with previous studies [29,32]. The aforementioned clinical symptoms were accompanied by a significant increase in the IHNV load in the skin and head kidney, reaching a peak at 7 dpi. Subsequently, the IHNV load gradually decreased, indicating that the immune system of rainbow trout fought against the invading IHNV and gradually eliminated the virus in the body during this process. Moreover, obvious IHNV fluorescent signals were detected in the epidermis, dermis, and even the muscle layer of rainbow trout skin. These results confirmed that the IHNV infection model in trout was successful, and IHNV could arrive at the skin mucosal site. Previous studies have indicated that IHNV invasion resulted in necrotic lesions in its target organs, including the spleen and head kidney [24]. However, very little is known regarding the pathological changes of fish mucosal tissues in response to viral infection. We observed severe tissue damage, inflammation, thinning of epidermal layer thickness, and a significant reduction in mucus cell numbers in the skin after IHNV infection, which might be caused by the necrosis and exfoliation of epithelial cells. The specific mechanism needs to be further explored. At the same time, we found that the significant pathological changes mainly occurred in the period of high IHNV load, indicating that not only was the virus gradually cleared, but also the body damage was gradually repaired in the late infection period. Since death after infection mainly occurs in the first two weeks, we can infer that surviving individuals have relatively stronger immunity, which is sufficient to successfully resist the invading virus, perhaps due to the large differences between different individuals. Additionally, we detected significant upregulation of antiviral genes in the skin of 4 and 7 dpi fish, including *mx1*, *ddx58*, *ifih1*, *stat1*, *dhx58*, and *tlr3*, suggesting that the antiviral immune responses in the fish skin were activated at the early stage of viral infection. Consistent with viral load and pathological changes, the intensity of the immune response was low around 21 days after infection. IHNV invasion also resulted in significant upregulation in the expression of *il6*, which has been shown to be essential for inflammatory resolution and migration and phagocytic activities of macrophages in mammals during viral infections [33]. Furthermore, *cath1*, which encodes an antibacterial peptide with broad-spectrum antibacterial activity [34], showed high expression at 14 dpi compared with other time points. Similarly, immersion infection with spring viraemia of carp virus (SVCV) induced high expression of *hepcidin* in the nose and pharynx mucosa of common carp [35]. This result reminds us that IHNV led to strong antiviral and inflammatory responses and may further result in antibacterial immune responses.

To further reveal the immune response and related pathways, we conducted transcriptome sequencing of skin samples from 4 and 14 dpi fish. Consistent with our qPCR results, antiviral genes were significantly upregulated in the early stage of infection (4 dpi), including *vig1*, *mx2*, *trim39*, *ifnar2*, *gig2e*, *ddx58*, *stat1b* and *dhx58*. Furthermore, KEGG enrichment analysis demonstrated that several pattern recognition receptor (PRR) signaling pathways were activated in the skin at 4 dpi, including the NOD-like receptor signaling pathway, C-type lectin receptor signaling pathway, and Toll-like receptor signaling pathway. Pattern recognition receptors are believed to recognize the conserved molecular structure of pathogens, thereby triggering the body’s innate immune response and playing a pivotal role against pathogenic infection [36,37]. Type I interferon response, an important interferon-mediated pathway in antiviral immunity, was also significantly enriched in GO analysis at 4 dpi. Most notably, we found that the expression of antibacterial genes in the 14 dpi group was dramatically increased compared with the 4 dpi group, among which the most representative genes were *hamp* (hepcidin antimicrobial peptide), *saa5* (serum amyloid A-5), *cd209* (CD209 molecule), *lyz2* (lysozyme type II), and *cath* (cathelicidin antimicrobial peptide). Hepcidins and cathelicidins are two major groups of natural antimicrobial peptides, which are important in the fight against bacterial invasion [38,39]. Lysozymes are key proteins to teleost that possess high bactericidal properties [40,41]. Several studies have reported that the expression of serum amyloid A-5 protein is significantly upregulated during bacterial infection. Moreover, the expression of *saa* was activated significantly earlier after *Yersinia ruckeri* reinfection in rainbow trout compared to primary infection [42]. As for CD209, it is a member of the C-type lectin family and is involved in the regulation of innate and adaptive immune system [43]. In addition, it has been reported that the expression of a CD209 homologue (CsCD209) of the teleost fish tongue sole (*Cynoglossus semilaevis*) was significantly upregulated during bacterial infection. Recombinant CsCD209 showed apparent binding abilities to a wide range of bacteria and fungi and significantly promoted the phagocytosis of bacteria by *C. semilaevis* leukocytes [44]. Consistent with the results of gene expression analysis, gene pathways associated with responses to bacterium were significantly enriched in the 14 dpi group according to GO pathway analysis. Collectively, our findings suggest that IHNV invasion activated antiviral innate immunity during the acute infection phase after infecting the skin of rainbow trout and then induced strong antibacterial response at the later stage of the infection. This likely occurred because IHNV destroyed the normal microbial structure of the skin, thus leading to the expansion of pathogenic bacteria.

Next, we assessed the changes in the diversity and composition of the microbial community of the fish skin at 4 and 14 dpi. Our results indicated that the richness and diversity of the skin microbiota significantly increased after viral infection, which was consistent with previous studies in fish infected with parasites and pathogenic bacteria. For instance, increased microbial diversity was observed in the skin microbiota of European seabass (*Dicentrarchus labrax*) infected with *Vibrio harveyi* and rainbow trout infected with *Ichthyophthirius multifiliis* [4,19]. This suggests that the effects of different pathogens on the microbial structure of fish skin are somewhat conserved. Additionally, non-metric multidimensional scaling (NMDS) combined with Anosim and Permanova analysis showed that the samples of 14 dpi group were distinctly separated from the control group, and the 4 dpi group was between them. This indicates that the microbial changes in the 14 dpi group may be more significant. Previous studies have demonstrated that Proteobacteria, Firmicutes, Bacteroidetes, and Actinobacteria make up the majority of the skin microbial community of aquatic animals [45,46]. Upon IHNV infection, the abundance of the predominant phylum Proteobacteria was substantially reduced, whereas that of Actinobacteria increased significantly. LEfSe (linear discriminant analysis effect size) analysis showed that the abundance of *Flavobacterium* increased significantly in 4 dpi fish. Many species of *Flavobacterium* are known to cause diseases in freshwater fish such as bacterial cold-water disease (BCWD) and columnaris disease, which are caused by *F. psychrophilum* and *F. columnare*, respectively [47,48]. A previous study on the skin microbial community of rainbow trout infected with *Ichthyophthirius multifiliis* also detected considerable increases in *Flavobacterium* in the 7 dpi group, suggesting that *Flavobacterium* could potentially be used as a biomarker of skin microbiota dysbiosis in rainbow trout following viral and parasitic invasion. In addition to the aforementioned changes in *Flavobacterium* abundance, the 4 dpi group also showed an increased abundance of *Chryseobacterium*. There has been an increasing number of cases of fish infections associated with different species of *Chryseobacterium* [49]. For example, *Chryseobacterium piscicola* has been linked to mortalities in farmed Atlantic salmon (*Salmo salar*) and rainbow trout (*Oncorhynchus mykiss*) [50,51]. Similarly, we found that the dominant genera in the 14 dpi group were mostly opportunistic pathogens. Concretely, we observed significant increases in the abundance of *Vibrio* and *Rhodococcus*, which became the biomarkers of the 14 dpi group. Many species of *Vibrio* are important pathogens in aquaculture responsible for vibriosis outbreaks, including *V. parahaemolyticus*, *V. harveyi*, and *V. owensii* [52]. Moreover, it has been reported that *Vibrio metschnikovii* was a potential pathogen of freshwater-cultured hybrid sturgeon [53], and our results showed that the abundance of *Vibrio metschnikovii* was significantly increased in the 14 dpi group. In the genus *Rhodococcus*, there are species that have been reported to be pathogenic to fish [54,55], as well as other species that have not been identified but still cause chronic granulomatous nephritis [56] and panophthalmitis in salmon [57]. Moreover, we detected a significant upregulation in the abundance of *Pseudomonas* (Pseudomonadaceae). The members of the *Pseudomonas* genus (Pseudomonadaceae) are part of the normal microbiota of fish but become highly opportunistic and pathogenic under stressful conditions such as overcrowding and pathogen infection, causing ulcerative syndrome and hemorrhagic septicemia [58]. Additionally, increased abundances of several other opportunistic pathogens (*Pseudoalteromonas* and *Uruburuella*) were also detected in the 14 dpi group. Researchers have previously revealed that invading viruses can enhance the expression of pathogenic bacterial receptors on the surface of respiratory epithelial cells, thereby promoting the adhesion of pathogenic bacteria to the surface of epithelial cells [59]. Therefore, we speculate that IHNV invasion may also increase the expression of receptors of some opportunistic pathogens. However, additional studies are needed to determine whether such a mechanism exists in teleost fish. In summary, IHNV infection affected the composition of the skin microbiota of rainbow trout and facilitated the growth of opportunistic pathogens, thus resulting in skin microbiota dysbiosis. This also explains the significant upregulation of antibacterial genes at 14 dpi.

In conclusion, the present study revealed the effects of viral infection on skin immune responses and mucosal microorganisms in rainbow trout. IHNV invasion caused shedding of epithelial cells in the skin, resulting in epidermal lesions. Transcriptome sequencing indicated that IHNV invasion could induce dramatic immune responses in trout skin. Several PRR pathways were activated, and antiviral genes were significantly upregulated in the early stage of infection, suggesting that the innate immune response played a crucial role in resisting viral invasion. In addition, the abundance of the predominant phylum Proteobacteria in normal trout significantly decreased after viral infection, whereas the abundances of various opportunistic bacteria increased significantly, leading to secondary bacterial infection. This corresponded with a concomitant upregulation of several antibacterial genes, suggesting that antibacterial pathways were activated due to secondary bacterial infection. Additionally, a previous study reported that some microorganisms on the skin mucosa are coated by the humoral immune molecule IgT [2]. Therefore, future studies should explore the changes in the adaptive humoral immune response caused by IHNV infection in the skin to gain a deeper understanding of the interactions between IgT, invasive pathogens, and skin microbiota. Overall, our study provides new insights into the effects of viral infection on the mucosal immune response and the microbial community of teleost skin.

## 4. Materials and Methods

### 4.1. Fish Maintenance

Juvenile rainbow trout (average weight 5.66 ± 0.89 g (mean ± SD)) used in this study were purchased from a fish farm in Chengdu (Sichuan province, China) and were transferred to the Aquaculture Base of College of Fisheries, Huazhong Agricultural University. After transport, fish were randomly distributed into the closed water-recirculation system with a temperature-controlled device for at least two weeks to acclimate to the laboratory conditions prior to experiments. Fish were acclimatized at 16 °C and fed with commercial diets at 1% of their body weight every day. Feeding was discontinued two days prior to the trials. Afterward, the fish were fed normally throughout the infection, except the day before sampling.

### 4.2. Cell Culture and IHNV Enrichment

For IHNV propagation, Epithelioma papulosum cyprini (EPC) cells were cultured in Minimum Essential Medium with Earl’s salts (MEM, Gibco, Gaithersburg, MD, USA) supplemented with 10% fetal bovine serum (FBS, Gibco, Amarillo, TX, USA) and 1% penicillin–streptomycin solution in 5% CO_2_ at 28 °C. EPC cell monolayers grown in T-75 flasks were infected with 10 μL IHNV. Then, the infected cells were grown in MEM containing 2% FBS at 18 °C and were observed for the development of virus-induced cytopathic effect (CPE) every day. When CPE was greater than 80%, the cells and supernatant were harvested. Intracellular viruses were collected using the repeated freeze-thaw method. As for the viral titer, viral supernatant was made into 10-fold serial dilutions (10^−1^–10^−10^), and then each dilution was added to six replicated wells in a 96-well plate bespread with EPC cells. The 50% tissue culture infectious dose (TCID50) was calculated by the statistical method of Reed and Meunch [60]. Titers were expressed as plaque-forming unit (PFU)/mL with one infectious unit equal to 0.7 TCID50. Subsequently, for the infection experiment, viral suspension was diluted to 1 × 10^9^ PFU·mL^−1^ in MEM, stored at −80 °C until use.

### 4.3. Sampling and Infection

For mimicking the natural route of infection, fish were bathed in 10 L aeration water containing 6 mL IHNV (1 × 10^9^ pfu·mL^−1^) for 2 h at 16 °C, and then, the fish were transferred into new, oxygenated tanks. Control fish were exposed to the same protocol with the exception of the absence of IHNV in the water. The fish were observed daily for disease symptoms and mortality for 30 days. For sampling, fish were anesthetized with MS-222 (final concentration, 160 mg·L^−1^). Nine fish were anesthetized and sampled on days 0, 1, 4, 7, 14, 21 and 28 after IHNV infection. When sampling, we stored 6 individual samples for RNA extraction, 6 individual samples for paraffin embedding, 3 individual samples for RNA-Seq, and 8 individual samples for 16S rRNA sequencing. For histological analysis, a piece of skin with muscle (~0.25 cm^2^) below the dorsal fin and above the lateral line was taken and immediately fixed in 4% paraformaldehyde neutral buffer solution at 4 °C for at least 24 h for paraffin embedding. For RNA extraction and Quantitative Real-time PCR, the skin (below the dorsal fin and above the lateral line, ~100 mg) and head kidney were removed and placed in the sterile micro-centrifuge tubes. For RNA-Seq and 16S rRNA sequencing, skin pieces (~100 mg) from above the lateral line and below the dorsal fin were excised and placed in the sterile micro-centrifuge tubes. All the samples collected for RNA or 16S rRNA sequencing were frozen promptly in liquid nitrogen and stored at −80 °C until use.

### 4.4. Histology, Light Microscopy, and Immunofluorescence Microscopy Studies

After fixed in 4% neutral formalin buffer, skin tissue sections at different time points were prepared by dehydration with gradient alcohol, paraffin embedding, and pathological section. Thereafter, the paraffin sections were stained with hematoxylin–eosin (H&E) and Alcian blue (AB) for structural and histological analysis. Images were acquired using an optical light microscope (Olympus, BX53, Shinjuku City, Tokyo, Japan) equipped with AxioVision software. The thickness and length of the epidermal layer of the skin were measured using the cellSens Standard software (Olympus, Shinjuku City, Tokyo, Japan). The number of mucus cells in the epidermal layer was counted by Adobe PhotoShop (Adobe PhotoShop 2020.Ink, San Jose, CA, USA). All results were measured blindly and independently by at least two researchers. According to the following procedure, the detection of IHNV infected cells in the skin was performed by immunofluorescence microscopy (Olympus, BX53, Shinjuku City, Tokyo, Japan). Briefly, sections were blocked for 30 min at room temperature (RT) with StartingBlock™ Blocking Buffer (Thermo Fisher Scientific, Waltham, MA, USA). Then, sections were incubated with mouse anti-IHNV-N mAb (Mouse IgG for isotype; 1 μg·mL^−1^; BIO−X Diagnostics, Rochefort, Belgium) at 4 °C overnight, followed by washes, incubation with Cy3-conjugated AffiniPure goat anti-mouse IgG pAb (3 μg·mL^−1^) at RT for 40 min, and counterstaining with DAPI (4′, 6-diamidino-2 phenylindole; 1 μg·mL^−1^; Invitrogen, Carlsbad, CA, USA) for 8 min before mounting. As controls, the mouse IgG was used as the primary antibody. The fluorescence images were merged using the Image color-merge channels function of the cellSens Standard software (Olympus, BX53, Shinjuku City, Tokyo, Japan).

### 4.5. RNA Extraction, cDNA Synthesis and RT-qPCR Analysis

Total RNA from skin and head kidney (~100 mg, n = 6) was extracted with TRIzol reagent (Invitrogen, Carlsbad, CA, USA) following the workflow outlined by the manufacturer and dissolved in DEPC-treated water. High purity of RNA was confirmed by spectrophotometric measurements (Implen NanoPhotometer NP 80 Touch, Munich, Germany) and RNA agarose gel electrophoresis (Agilent Bioanalyser, 2100). Then, 1 µg of total RNA was reverse transcribed in a 20 μL reaction volume using the Hifair III 1st Strand cDNA Synthesis SuperMix for qPCR (gDNA digester plus) (YEASEN, Shanghai, China). Subsequently, the resulting cDNA was diluted to 300 ng/μL for qPCR, which was performed using Hieff qPCR SYBR Green Master Mix (YEASEN, Shanghai, China) according to the manufacturer’s instructions. The qPCR amplification procedure was as follows: 95 °C for 5 min, followed by 41 cycles of 95 °C for 10 s and 58 °C for 30 s. Sequence information for qPCR primers is summarized in Appendix A. After evaluating the performance of the housekeeping genes EF1α and β-actin, we found that they were both stable in the experiments. In the subsequent qPCR analyses, we selected EF1α as the internal reference gene for normalization. Average fold change was determined using the 2^−ΔΔCt^ method [61]. A standard curve of IHNV plasmid DNA was used to calculate the viral copy number in samples by applying their cycle threshold (Ct) values. The standard curve is shown in Appendix A.

### 4.6. Plaque Assay of IHNV

The skin samples (below the dorsal fin and above the lateral line) from control and infected groups were cut into small pieces (~0.1 cm^2^) and were mechanically disaggregated on a 100 μm cell strainer with DMEM, and then, the cell suspension was collected. All cell suspensions were then freeze-thawed three times to release intracellular virus particles after adding 5% FBS. The cell suspension was centrifuged for 5 min at 400× *g* once and 10,000× *g* thrice at 4 °C to remove cell debris, and then, the supernatant was filtered with a 0.45 μm membrane filter to remove bacteria in the mucosal tissue. EPC cells were seeded into a 6-well plate at a density of 1 × 10^6^ cells per well and incubated at 28 °C for 24 h in advance. Then, the viral supernatant (50 μL) was inoculated to the individual well of the plate. When CPEs were clearly apparent, images of infected cells were captured using a microscope.

### 4.7. RNA-Seq Library Construction and Transcriptome Analysis

Skin samples from control and infected groups were sent to Shanghai Personal Biotechnology Co., Ltd. (Shanghai, China) for RNA-seq library construction and transcriptome analysis. Briefly, total RNA from skin (~100 mg, 3 samples each from the control, 4 dpi and 14 dpi groups) was extracted using the Trizol reagent (Invitrogen Life Technologies, Carlsbad, CA, USA), and then, the quantification and quality were assured by NanoDrop spectrophotometer (Thermo Scientific, Waltham, MA, USA). The following steps were performed to generate the RNA-seq library for sequencing. The poly(A)-containing mRNA molecules were purified using poly-T oligo-attached magnetic beads. Fragmentation was carried out using divalent cations under elevated temperature in Illumina proprietary fragmentation buffer. The first-strand cDNA was synthesized using random oligonucleotides and Super Script II, and the second-strand cDNA was then synthesized using DNA Polymerase I and RNase H. Double-stranded cDNA fragments were purified for end repair, followed by 3′ adenylation, and the ligation of sequencing adaptors. Size-selection and purification of cDNA fragments of approximately 400–500 bp in length were performed using AMPure XP beads (Beckman-Coulter, Indianapolis, IN, USA). PCR amplification was performed, and the PCR products were purified again with AMPure XP beads, and finally, the library was obtained. The sequencing library was then paired-end sequenced (2 × 150 bp) with average sequencing depth of 49,072,519 reads on the NovaSeq 6000 platform (Illumina).

The sequenced raw reads were processed to obtain high-quality, clean reads using the following strategy: (i) remove reads with adapter contamination; (ii) remove low quality reads whose average quality was less than Q20. Subsequently, the filtered reads were mapped to the *Oncorhynchus mykiss* genome (GCF_013265735.2_USDA_OmykA_1.1_genomic.fna) using HISAT2 v2.0.5. Gene readcounts were estimated with HTSeq (0.9.1) statistics and expression levels of genes were normalized using FPKM values. Differential expression analysis was performed using the DESeq software (1.30.0). The conditions for screening differentially expressed genes (DEGs) were as follows: |log2 (FoldChange)| > 1, *p* (adj) < 0.05. Gene Ontology (GO) functional enrichment analysis and Kyoto Encyclopedia of Genes and Genomes (KEGG) pathway enrichment analysis were performed for understanding the pathway enrichment of DEGs after virus infection. GO enrichment analysis was performed using the Bioconductor package topGO. KEGG pathway enrichment analysis was performed using clusterProfiler (3.4.4) software. Raw RNA-Seq data have been deposited to the NCBI Sequence Read Archive (SRA) under BioProject accession number PRJNA871965.

### 4.8. DNA Amplification, 16S rRNA Sequencing and Bioinformatics Analysis

First, 16S rRNA gene amplification and sequencing were conducted on the Illumina NovaSeq platform at Shanghai Personal Biotechnology Co., Ltd. (Shanghai, China). Briefly, total bacterial DNA from skin samples (8 samples each from the control, 4 dpi and 14 dpi groups) was extracted using the OMEGA Soil DNA Kit (D5625-01) following the manufacturer’s instructions. The quantity of isolated DNA was measured using a NanoDrop NC2000 spectrophotometer (Thermo Scientific, Waltham, MA, USA), and the quality was checked by agarose gel (1.2%) electrophoresis. To evaluate the bacterial community composition, the forward primer (338F 5′- ACTCCTACGGGAGGCAGCA-3′) and the reverse primer (806R 5′-GGACTACHVGGGTWTCTAAT-3′) were used to amplify the V3–V4 regions of the 16S rRNA gene. Sample-specific 7 bp barcodes were introduced for multiplex sequencing using an Illumina MiSeq instrument. The PCR products were purified with Agencourt AMPure Beads (Beckman Coulter, Indianapolis, IN, USA) and quantified by using the PicoGreen dsDNA Assay Kit (Invitrogen, Carlsbad, CA, USA). Each sample was then diluted in TE buffer and pooled. The pooled sample was sequenced on the NovaSeq 6000 platform (Illumina) using the 2 × 250 bp paired-end protocol, with ~80,000 reads generated per sample. Raw reads were demultiplexed with the “demux” plugin and quality filtered, trimmed, merged, denoised, chimera filtered using the DADA2 plugin. Then, the remaining high-quality sequences were clustered into amplicon sequence variants (ASVs) (equivalent to 100% similarity operational taxonomic units (OTU) in the conventional practice). Bioinformatics processing was performed with QIIME2 and R packages. Taxonomy assignment was performed using QIIME2 against the Greengenes Database (Release 13.8, http://greengenes.secondgenome.com/, accessed on 22 October 2022) [62]. Alpha diversity and Beta diversity analysis (using the ggplot2 and vegan packages in R) was conducted to compare the richness and evenness of ASVs among samples and the structural variation of microbial communities across samples, respectively. Taxa relative abundances at the phylum, class, order, family, genus levels were statistically compared among groups by Metastats [63]. Clustering heatmap analysis was conducted using the pheatmap package in R. LEfSe (linear discriminant analysis effect size) (using the ggtree package in R) was performed to detect differentially abundant taxa across groups using the default parameters (The LDA threshold is 2). Raw 16S rRNA sequencing data have been deposited to the NCBI Sequence Read Archive (SRA) under BioProject accession number PRJNA872024.

### 4.9. Statistical Analysis

For the comparison of survival curves (Kaplan–Meier curves) in control and IHNV-infected groups, the Log-rank (Mantel–Cox) test was used (version 6.01; GraphPad Prism). When comparing the thickness of epidermal layer and the number of mucus cells in skin at different time points after infection, statistical differences were evaluated by unpaired Student’s *t* test (GraphPad Prism, San Diego, CA, USA; version 6.01). Changes in gene expression at different time points after infection were compared with unpaired Student’s *t* test (GraphPad Prism).

Kruskal–Wallis test and Dunn’s post hoc test were used to verify the significance of differences in alpha diversity between different groups. Significance of differences between sample clusters in NMDS analysis were assessed with Anosim (Analysis of similarities) and Permanova (Permutation multivariate analysis of variance) with the vegan package of R. The relative abundance of specific bacterial taxa in different groups were compared using the Kruskall–Wallis test followed by Mann–Whitney test (GraphPad Prism). The nonparametric Kruskal–Wallis and Wilcoxon rank sum test were used for microbiota LEfSe enrichment analysis (LEfSe package of Python). All data are expressed as mean ± SEM.

## Figures and Tables

**Figure 1 ijms-23-14037-f001:**
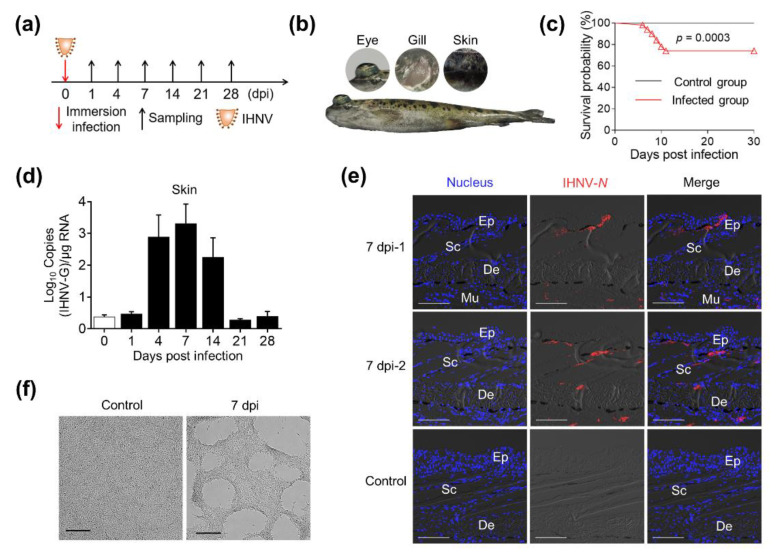
Successful invasion of IHNV into rainbow trout after immersion infection. (**a**) A schematic diagram about the timing of infection and sampling. (**b**) Fish mortalities and clinical symptoms were recorded daily for 30 days post IHNV infection. Symptoms of exophthalmia, pale gills, and darkening of the skin appeared in diseased fish 4 to 7 days post infection. (**c**) Cumulative survival rates of control and IHNV-infected group. Group comparisons for survival data were made by log-rank test statistics. (**d**) IHNV-G gene viral load (log10) at different time points was assessed in skin samples using a RT-qPCR assay. (**e**) Immunofluorescence staining of IHNV in skin paraffin sections from control and infected fish (7 dpi) (n = 6). IHNV-positive cells (red) were stained with an anti-IHNV-N mAb; nuclei were stained with DAPI (blue). Ep, epidermis; Sc, scale; De, dermis. Scale bars, 100 μm. (**f**) Cytopathic effects on EPC cells after incubation with skin homogenate supernatant from control fish and 7 dpi fish. Scale bars, 100 μm.

**Figure 2 ijms-23-14037-f002:**
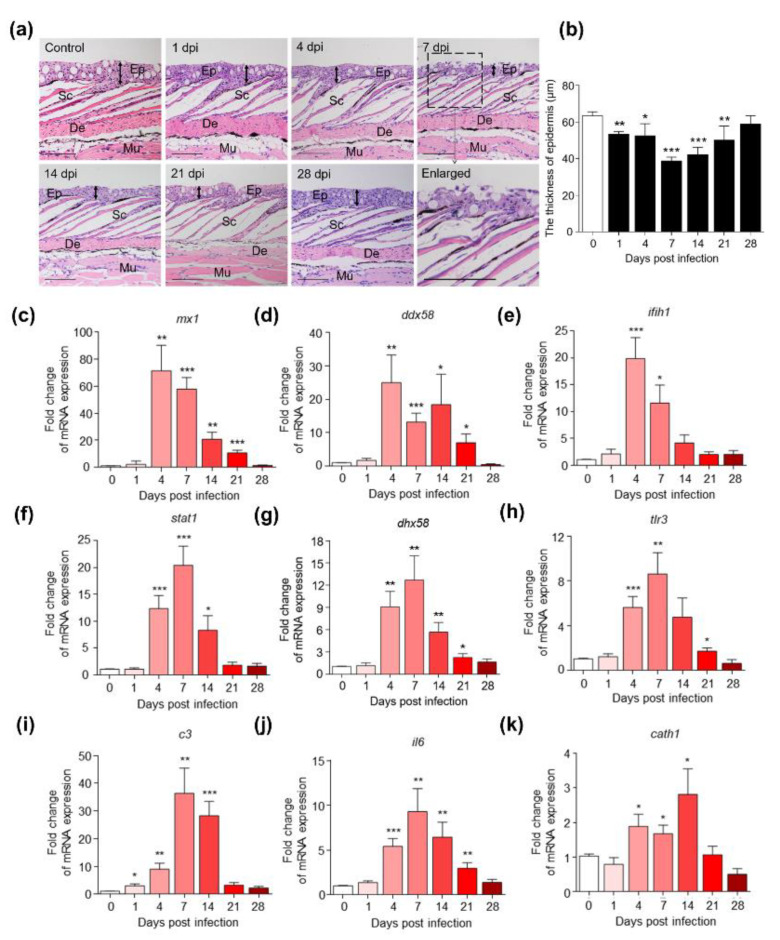
Histopathological changes and gene expression changes in rainbow trout skin after IHNV invasion. (**a**) Hematoxylin–eosin (H&E) staining of skin paraffin sections at the indicated time points. Ep, epidermis; Sc, scale; De, dermis. Scale bars, 100 μm. Black two-way arrows represent the location of the epidermal thickness measurement. (**b**) Histogram showing the thickness of skin epidermis at the indicated time points (n = 6). (**c**–**k**) Histogram comparing the expression of 9 immune-related genes (mx1, ddx58, ifih1, stat1, dhx58, tlr3, c3, il6, cath1) at indicated different time points following IHNV infection. In all cases, the expression at 0 dpi was taken as a calibrator against which the relative levels of other time points were calculated. * *p* < 0.05, ** *p* < 0.01, *** *p* < 0.001.

**Figure 3 ijms-23-14037-f003:**
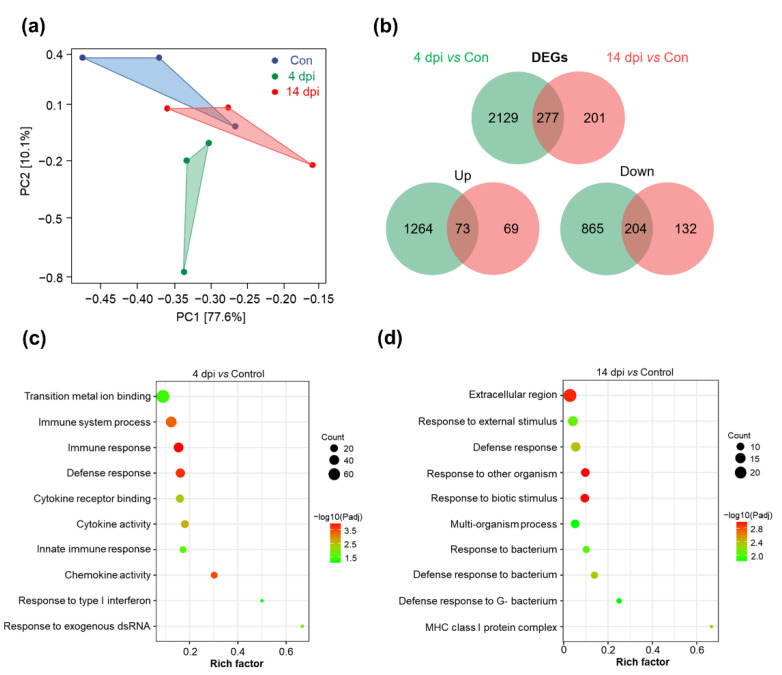
Analysis of the differentially expressed genes (DEGs) in the 4 and 14 dpi groups compared with the control group. (**a**) Principal component analysis (PCA) plot shows clustering of three groups, with axes corresponding to the two different principal components. The blue, green and red dots represent the control, 4 dpi and 14 dpi samples, respectively, with 3 parallel samples in each group. (**b**) Venn diagrams of RNA-Seq showing the overlap of genes upregulated or downregulated in the skin at 4 or 14 dpi versus control fish. (**c**,**d**) GO enrichment analysis of DEGs related to defense responses and immune responses. The size of the bubbles represents number of genes in a pathway, while the color represents −Log10(*p*-adj). Rich factor of the X-axis refers to (the number of DEGs annotated to the GO term)/(the total number of genes annotated to the GO term).

**Figure 4 ijms-23-14037-f004:**
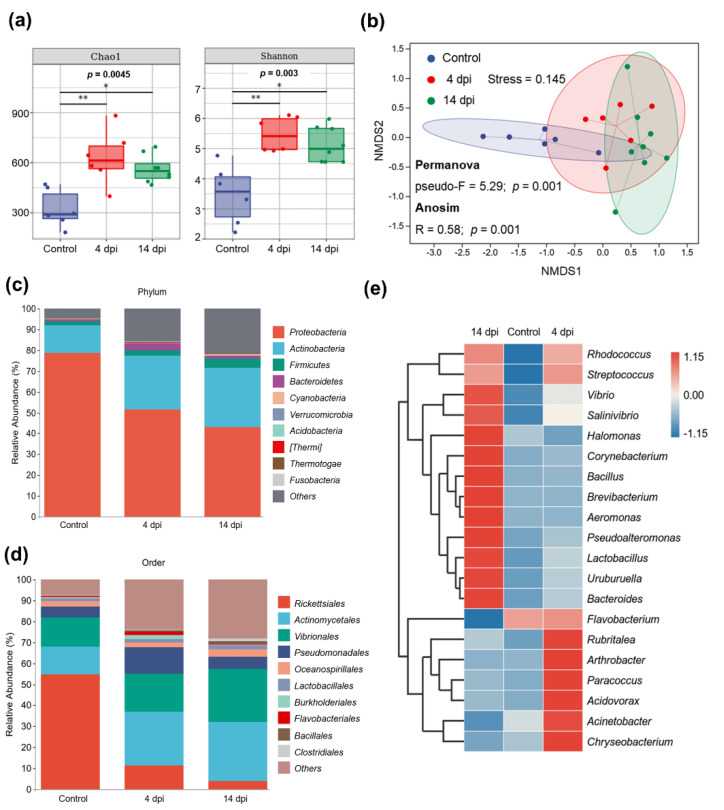
IHNV infection induced the changes in skin microbial diversity and composition. (**a**) Chao1 index and Shannon’s diversity index were used to measure alpha diversity between control and infection group (control group, n = 6; 4 dpi group, n = 6; 14 dpi group, n = 8). The number under the diversity index label is the *p* value of the Kruskal–Wallis test, and the significance of Dunn’s post hoc test was marked: * *p* < 0.05, ** *p* < 0.01. (**b**) Non-metric multidimensional scaling (NMDS) plot (Stress = 0.145) based on Bray–Curtis distance demonstrating the differences in microbial community composition. The blue, red and green dots represent the control, 4 dpi and 14 dpi samples, respectively. The ellipse represents the 95% confidence interval. The microbial structure differed significantly between groups confirmed by Permanova (Pseudo-F = 5.29; *p* = 0.001) and Anosim (R = 0.58; *p*  <  0.001). (**c**,**d**) Stacked bar chart at the phylum (**c**) and order (**d**) level showing the average relative abundance of the bacterial community composition for samples from control, 4 dpi and 14 dpi groups (Top 10 taxa), with different taxa shown as different colors. Data derive from the average value from each group. (**e**) Heatmap and cluster analysis showing the average taxonomic composition of the top 20 most abundant genera of the three groups. The color represents the relative abundance of the corresponding genus, with red indicating the higher abundance and blue indicating the lower abundance (key at right). (**c**–**e**): Significant differences of different taxa were determined by non-parametric analysis using the Kruskall–Wallis test followed by Mann–Whitney test. The statistical data are shown in Appendix A.

**Figure 5 ijms-23-14037-f005:**
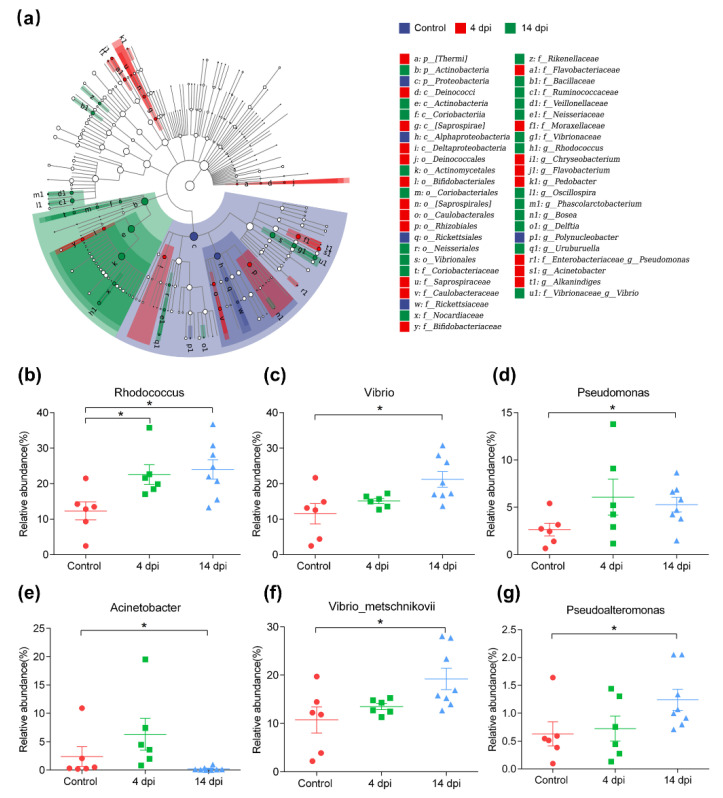
Characteristics of microbial community composition of skin in control and infected fish. (**a**) Cladogram generated from linear discriminant analysis effect size (LEfSe) analysis showing the most differentially abundant taxa enriched in microbiota from control group (blue), 4 dpi group (red) and 14 dpi group (green). The central point represents the root of the tree (Bacteria), and each ring represents the next lower taxonomic level (kingdom to genus: *p*, phylum; c, class; o, order; f, family; g, genus). Solid circles indicate significant differences. The diameter of each circle represents the relative abundance of the taxon. The LDA threshold is 2. (**b**–**g**) Scatter plots showing some taxa with significant changes in relative abundance between control and infected group: Rhodococcus (**b**), Vibrio (**c**), Pseudomonas (**d**), Acinetobacter (**e**), Vibrio metschnikovii (**f**), Pseudoalteromonas (**g**). * *p* < 0.05, significantly different by Mann–Whitney U test.

**Table 1 ijms-23-14037-t001:** Expression changes of 20 representative immune-related genes.

Gene Name	Accession Number	Description	4 dpi vs. Control			14 dpi vs. Control		
			Fpkm (vs. Control)	FC	*p* (adj)	Fpkm (vs. Control)	FC	*p* (adj)
*hamp*	XM_021595153.2	hepcidin	30.77 ± 5.72	42.18	9.01 × 10^−5^	690.8 ± 379.0	717.01	0.031
*saa5*	XM_021586448.2	serum amyloid A-5 protein	44.86 ± 17.66	53.91	6.88 × 10^−4^	470.7 ± 262.7	468.26	0.038
*cd209*	NM_001124633.1	CD209 molecule	11.72 ± 2.31	14.82	2.90 × 10^−3^	88.64 ± 27.78	88.42	2.38 × 10^−4^
*cath*	NM_001124463.1	cathelicidin antimicrobial peptide	4.27 ± 0.81	5.15	3.79 × 10^−5^	14.37 ± 4.37	13.20	0.002
*lyz2*	NM_001124716.1	lysozyme II	4.81 ± 2.63	5.69	0.343 ^ns^	15.51 ± 2.96	15.38	3.73 × 10^−5^
*vig1*	NM_001124253.1	viperin	345.20 ± 39.88	430.66	1.47 × 10^−43^	85.26 ± 65.47	85.83	0.366 ^ns^
*mx2*	NM_001124751.1	interferon-induced GTP-binding protein Mx2	83.21 ± 13.71	101.76	2.55 × 10^−31^	14.58 ± 6.55	14.27	0.208 ^ns^
*trim39*	XM_021573556.2	E3 ubiquitin-protein ligase TRIM39-like	74.67 ± 1.67	96.74	1.21 × 10^−15^	20.68 ± 5.72	20.90	5.14 × 10^−5^
*ifnar2*	XM_021578985.2	interferon alpha/beta receptor 2-like	62.86 ± 11.18	76.09	5.39 × 10^−28^	22.2 ± 9.6	21.53	0.111 ^ns^
*irf7*	XM_021600499.2	interferon regulatory factor 7	56.75 ± 12.38	71.40	2.89 × 10^−19^	24.63 ± 12.14	24.91	0.190 ^ns^
*irf8*	XM_036956322.1	interferon regulatory factor 8	37.54 ± 2.30	48.23	6.93 × 10^−24^	8.7 ± 4.88	8.80	0.464 ^ns^
*gig2e*	XM_036944105.1	grass carp reovirus (GCRV)-induced gene 2e	40.28 ± 15.07	46.59	2.25 × 10^−4^	11.12 ± 4.62	10.58	0.205 ^ns^
*ddx58*	XM_036973405.1	DEAD (Asp-Glu-Ala-Asp) box polypeptide 58	25.37 ± 0.80	31.83	7.00 × 10^−21^	7.67 ± 3.25	7.51	0.289 ^ns^
*stat1b*	XM_021579196.2	signal transducer and activator of transcription 1b	12.61 ± 0.39	16.13	1.34 × 10^−14^	3.81 ± 1.05	3.75	0.207 ^ns^
*dhx58*	XM_036938249.1	DEXH (Asp-Glu-X-His) box polypeptide 58	10.06 ± 1.49	13.00	1.76 × 10^−5^	2.69 ± 1.53	2.66	0.761 ^ns^
*ccl19a.1*	XM_021605155.2	C-C motif chemokine 19a.1	174.20 ± 47.88	221.68	5.49 × 10^−14^	182.3 ± 135.1	188.58	0.274 ^ns^
*tlr3*	NM_001124578.1	toll-like receptor 3	6.45 ± 0.82	8.08	7.58 × 10^−9^	3.7 ± 0.71	3.62	0.039
*il4i1*	XM_036961927.1	interleukin 4 induced 1	56.01 ± 7.00	69.92	9.62 × 10^−27^	23.91 ± 14.47	23.75	0.348 ^ns^
*c1q1*	XM_021624859.2	complement C1q-like protein 2	12.45 ± 4.76	15.30	0.004	137.5 ± 58.09	135.74	0.008
*c4b*	NM_001124385.1	complement C4-B precursor	44.67 ± 21.22	56.24	0.011	31.71 ± 11.96	29.90	0.018

Fpkm (vs. control) represents the expression change of the gene of interest in the infected group relative to the control group, it is expressed as mean ± SEM; FC, fold change; If “ns” is marked at the top right of *p* (adj), it means that there is no significant difference in the expression of this gene in the corresponding group.

**Table 2 ijms-23-14037-t002:** KEGG enrichment analysis of DEGs in skin at 4 days post IHNV infection.

Category	Pathway Terms	Number of DEGs			*p* (adj)
		Up	Down	Total	
Immune system	NOD-like receptor signaling pathway	38	6	44	0.01590
	C-type lectin receptor signaling pathway	17	14	31	0.01590
	Toll-like receptor signaling pathway	21	8	29	0.03047
Signaling molecules and interaction	Cytokine-cytokine receptor interaction	44	15	59	0.00128
	ECM-receptor interaction	2	25	27	0.00506
Cellular community	Focal adhesion	8	43	51	0.03047
Cell growth and death	Necroptosis	40	1	41	0.02875
Folding, sorting and degradation	Protein processing in endoplasmic reticulum	28	11	39	0.02728
	Proteasome	34	0	34	6.43 × 10^−14^
Energy metabolism	Oxidative phosphorylation	35	0	35	0.00018

**Table 3 ijms-23-14037-t003:** KEGG enrichment analysis of DEGs in skin at 14 days post IHNV infection.

Category	Pathway Terms	Number of DEGs			*p* (adj)
		Up	Down	Total	
Signaling molecules and interaction	ECM-receptor interaction	1	17	18	2.72 × 10^−10^
Cellular community	Focal adhesion	0	23	23	3.01 × 10^−7^
Folding, sorting and degradation	Proteasome	7	0	7	0.00298

## Data Availability

The raw RNA-Seq data of this study have been deposited to the NCBI Sequence Read Archive (SRA) under BioProject accession number PRJNA871965. Raw 16S rRNA sequencing data have been deposited to the NCBI Sequence Read Archive (SRA) under BioProject accession number PRJNA872024.

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
