# Peer review of "Alterations of the Mucosal Immune Response and Microbial Community of the Skin upon Viral Infection in Rainbow Trout (Oncorhynchus mykiss)"

_ijms, 2022, doi:10.3390/ijms232214037_

Round 1
Reviewer 1 Report
Would you conclude that the fish at 21 dpi had successfully fought off the infection, and were recovering, resulting in the decreased immune response? As most of the deaths occurred in the first 2 weeks, did these survivors have better immune functioning? The high mortality (30%) early in infection indicates that there are differences in immune response between individuals, an interesting finding in itself.
Author Response
Dear Editor and Reviewers:
Thanks for your letter and for the constructive comments concerning our manuscript entitled “Alterations of the mucosal immune response and microbial community of the skin upon viral infection in rainbow trout (Oncorhynchus mykiss)”. At the same time, thank you very much for your approval of our research content and manuscript. We have made correction according to the comments, revised portion are marked in red in the paper. The responses to comments are as follows:
All the best,
Authors.
Reviewer1:Would you conclude that the fish at 21 dpi had successfully fought off the infection, and were recovering, resulting in the decreased immune response? As most of the deaths occurred in the first 2 weeks, did these survivors have better immune functioning? The high mortality (30%) early in infection indicates that there are differences in immune response between individuals, an interesting finding in itself.
Response: Yes, we totally agree with you! Because the IHNV load in rainbow trout continued to be high at 4~14 days after the invasion, then its load gradually decreased, and the IHNV was basically cleared by the immune system about 21 days after the invasion. Therefore, we infer that the fish about 21 days after infection has basically completely fought against the invading virus, and the damaged body gradually recovered. Therefore, the immune response is also low at 21 dpi. In addition, post-infection death mainly occurred in the first two weeks, so it can be assumed that the fish that survived afterward had a relatively stronger immune function, which was due to the large individual differences between different individuals. In addition, we discussed these contents in the discussion section. Please see the following sections of the revised manuscript for details:
Line 344-346: “Subsequently, the IHNV load gradually decreased, indicating that the immune system of rainbow trout fought against the invading IHNV and gradually eliminated the virus in the body during this process.”
Line 356-362: “At the same time, we found that the significant pathological changes mainly occurred in the period of high IHNV load, indicating that not only the virus was gradually cleared, but also the body damage was gradually repaired in the late infection period. Since death after infection mainly occurs in the first two weeks, we can infer that surviving individuals have relatively stronger immunity, which is sufficient to successfully resist the invading virus, perhaps due to the large differences between different individuals.”
Line 365-367: “Consistent with viral load and pathological changes, the intensity of the immune response was low around 21 days after infection.”
Reviewer 2 Report
The MS entitled “Alterations of the mucosal immune response and microbial community of the skin upon viral infection in rainbow trout (Oncorhynchus mykiss)” (IJMS 2016868) submitted by Zhan et al is interesting research about the innate immunity developed by INNV infection and also studied about the bacterial secondary infection
Abstract
Well addressed the secondary infection due to INHV infection and the immune gene expression
Introduction
Well written about the problems supported by the literature. The aim may include the last paragraph of the introduction
Results
Well organized and the data analysis is a good manner. The figure quality and presentation is very excellent
Discussion and Conclusion
They discussed all data with suitable supporting literature. The conclusion is very informative for the study.
Materials and methods
Well-written and standard protocols were followed
Line 576: Oncorhynchus mykis will be changed to O. mykis with italic style
After the minor revision, the MS may accept it for publication
Author Response
Dear Editor and Reviewers:
Thanks for your letter and for the constructive comments concerning our manuscript entitled “Alterations of the mucosal immune response and microbial community of the skin upon viral infection in rainbow trout (Oncorhynchus mykiss)”. At the same time, thank you very much for your approval of our research content and manuscript. We have made correction according to the comments, revised portion are marked in red in the paper. The responses to comments are as follows:
All the best,
Authors
Reviewer2:
Abstract
Well addressed the secondary infection due to INHV infection and the immune gene expression.
Response: Thanks for the comments.
Introduction
Well written about the problems supported by the literature. The aim may include the last paragraph of the introduction
Response: Thanks for the comments.
Results
Well organized and the data analysis is a good manner. The figure quality and presentation is very excellent
Response: Thanks for the comments.
Discussion and Conclusion
They discussed all data with suitable supporting literature. The conclusion is very informative for the study.
Response: Thanks for the comments.
Materials and methods
Well-written and standard protocols were followed
Line 576: Oncorhynchus mykis will be changed to O. mykis with italic style.
Response: Thank you for pointing out the mistake here. We have corrected it in the revised manuscript.
Line 603: “......were mapped to the Oncorhynchus mykiss genome...... ”